# The eukaryotic bell-shaped temporal rate of DNA replication origin firing emanates from a balance between origin activation and passivation

Jean-Michel Arbona[1], Arach Goldar[2], Olivier Hyrien[3], Alain Arneodo[4], Benjamin Audit[1]*

[1]Laboratoire de Physique, Université de Lyon, Ens de Lyon, Université Claude Bernard Lyon 1, CNRS , Lyon, France; [2]Ibitec-S, CEA, Gif-sur-Yvette, France; [3]Institut de Biologie de l'Ecole Normale Supérieure, Ecole Normale Supérieure, CNRS, INSERM, PSL Research University, Paris, France; [4]LOMA, Univ de Bordeaux, CNRS, UMR 5798, Talence, France

**Abstract** The time-dependent rate $I(t)$ of origin firing per length of unreplicated DNA presents a universal bell shape in eukaryotes that has been interpreted as the result of a complex time-evolving interaction between origins and limiting firing factors. Here, we show that a normal diffusion of replication fork components towards localized potential replication origins (*p-oris*) can more simply account for the $I(t)$ universal bell shape, as a consequence of a competition between the origin firing time and the time needed to replicate DNA separating two neighboring *p-oris*. We predict the $I(t)$ maximal value to be the product of the replication fork speed with the squared *p-ori* density. We show that this relation is robustly observed in simulations and in experimental data for several eukaryotes. Our work underlines that fork-component recycling and potential origins localization are sufficient spatial ingredients to explain the universality of DNA replication kinetics.
DOI: https://doi.org/10.7554/eLife.35192.001

*For correspondence:
benjamin.audit@ens-lyon.fr

## Introduction

Eukaryotic DNA replication is a stochastic process (*Hyrien et al., 2013*; *Hawkins et al., 2013*; *Hyrien, 2016b*). Prior to entering the S(ynthesis)-phase of the cell cycle, a number of DNA loci called potential origins (*p-oris*) are *licensed* for DNA replication initiation (*Machida et al., 2005*; *Hyrien et al., 2013*; *Hawkins et al., 2013*). During S-phase, in response to the presence of origin *firing* factors, pairs of replication *forks* performing bi-directional DNA synthesis will start from a subset of the *p-oris*, the active replication origins for that cell cycle (*Machida et al., 2005*; *Hyrien et al., 2013*; *Hawkins et al., 2013*). Note that the inactivation of *p-oris* by the passing of a replication fork called origin *passivation*, forbids origin firing in already replicated regions (*de Moura et al., 2010*; *Hyrien and Goldar, 2010*; *Yang et al., 2010*). The time-dependent rate of origin firing per length of unreplicated DNA, $I(t)$, is a fundamental parameter of DNA replication kinetics. $I(t)$ curves present a universal bell shape in eukaryotes (*Goldar et al., 2009*), increasing toward a maximum after mid-S-phase and decreasing to zero at the end of S-phase. An increasing $I(t)$ results in a tight dispersion of replication ending times, which provides a solution to the random completion problem (*Hyrien et al., 2003*; *Bechhoefer and Marshall, 2007*; *Yang and Bechhoefer, 2008*).

Models of replication in *Xenopus* embryo (*Goldar et al., 2008*; *Gauthier and Bechhoefer, 2009*) proposed that the initial $I(t)$ increase reflects the progressive import during S-phase of a limiting origin firing factor and its recycling after release upon forks merge. The $I(t)$ increase was also

**eLife digest** Before a cell can divide, it must duplicate its DNA. In eukaryotes – organisms such as animals and fungi, which store their DNA in the cell's nucleus – DNA replication starts at specific sites in the genome called replication origins. At each origin sits a protein complex that will activate when it randomly captures an activating protein that diffuses within the nucleus. Once a replication origin activates or "fires", the complex then splits into two new complexes that move away from each other as they duplicate the DNA. If an active complex collides with an inactive one at another origin, the latter is inactivated – a phenomenon known as origin passivation. When two active complexes meet, they release the activating proteins, which diffuse away and eventually activate other origins in unreplicated DNA.

The number of origins that activate each minute divided by the length of unreplicated DNA is referred to as the "rate of origin firing". In all eukaryotes, this rate – also known as I(t) – follows the same pattern. First, it increases until more than half of the DNA is duplicated. Then it decreases until everything is duplicated. This means that, if plotted out, the graph of origin firing rate would always be a bell-shaped curve, even for organisms with genomes of different sizes that have different numbers of origins. The reason for this universal shape remained unclear.

Scientists had tried to create numerical simulations that model the rate of origin firing. However, for these simulations to reproduce the bell-shape curve, a number of untested assumptions had to be made about how DNA replication takes place. In addition, these models ignored the fact that it takes time to replicate the DNA between origins.

To take this time into account, Arbona et al. instead decided to model the replication origins as discrete and distinct entities. This way of building the mathematical model succeeded in reproducing the universal bell curve shape without additional assumptions. With this simulation, the balance between origin activation and passivation is enough to achieve the observed pattern.

The new model also predicts that the maximum rate of origin firing is determined by the speed of DNA replication and the density of origins in the genome. Arbona et al. verified this prediction in yeast, fly, frog and human cells – organisms with different sized genomes that take between 20 minutes and 8 hours to replicate their DNA. Lastly, the prediction also held true in yeast treated with hydroxyurea, an anticancer drug that slows DNA replication.

A better understanding of DNA replication can help scientists to understand how this process is perturbed in cancers and how drugs that target DNA replication can treat these diseases. Future work will explore how the 3D organization of the genome affects the diffusion of activating proteins within the cell nucleus.

DOI: https://doi.org/10.7554/eLife.35192.002

reproduced in a simulation of human genome replication timing that used a constant number of firing factors having an increasing reactivity through S-phase (*Gindin et al., 2014*). In these three models, an additional mechanism was required to explain the final $I(t)$ decrease by either a subdiffusive motion of the firing factor (*Gauthier and Bechhoefer, 2009*), a dependency of firing factors' affinity for *p-oris* on replication fork density (*Goldar et al., 2008*), or an inhomogeneous firing probability profile (*Gindin et al., 2014*). Here, we show that when taking into account that *p-oris* are distributed at a finite number of localized sites then it is possible to reproduce the universal bell shape of the $I(t)$ curves without any additional hypotheses than recycling of fork components. $I(t)$ increases following an increase of fork mergers, each merger releasing a firing factor that was trapped on DNA. Then $I(t)$ decreases due to a competition between the time $t_c$ to fire an origin and the time $t_r$ to replicate DNA separating two neighboring *p-ori*. We will show that when $t_c$ becomes smaller than $t_r$, *p-ori* density over unreplicated DNA decreases, and so does $I(t)$. Modeling random localization of active origins in *Xenopus* embryo by assuming that every site is a (weak) *p-ori*, previous work implicitly assumed $t_r$ to be close to zero (*Goldar et al., 2008*; *Gauthier and Bechhoefer, 2009*) forbidding the observation of a decreasing $I(t)$. Licensing of a limited number of sites as *p-ori* thus appears to be a critical property contributing to the observed canceling of $I(t)$ at the end of S-phase in all studied eukaryotes.

## Results

### Emergence of a bell-shaped $I(t)$

In our modeling of replication kinetics, a bimolecular reaction between a diffusing firing factor and a *p-ori* results in an origin firing event; then each half of the diffusing element is trapped and travels with a replication fork until two converging forks merge (termination, *Figure 1a*). A molecular mechanism explaining the synchronous recruitment of firing factors to both replication forks was recently proposed (*Araki, 2016*), supporting the bimolecular scenario for *p-ori* activation. Under the assumption of a well-mixed system, for every time step $dt$, we consider each interaction between the $N_{FD}(t)$ free diffusing firing factors and the $N_{p-ori}(t)$ *p-oris* as potentially leading to a firing with a probability $k_{on}dt$. The resulting simulated firing rate per length of unreplicated DNA is then:

$$I_S(t) = \frac{N_{fired}(t, t+dt)}{L_{unrepDNA}(t)dt},$$

(1)

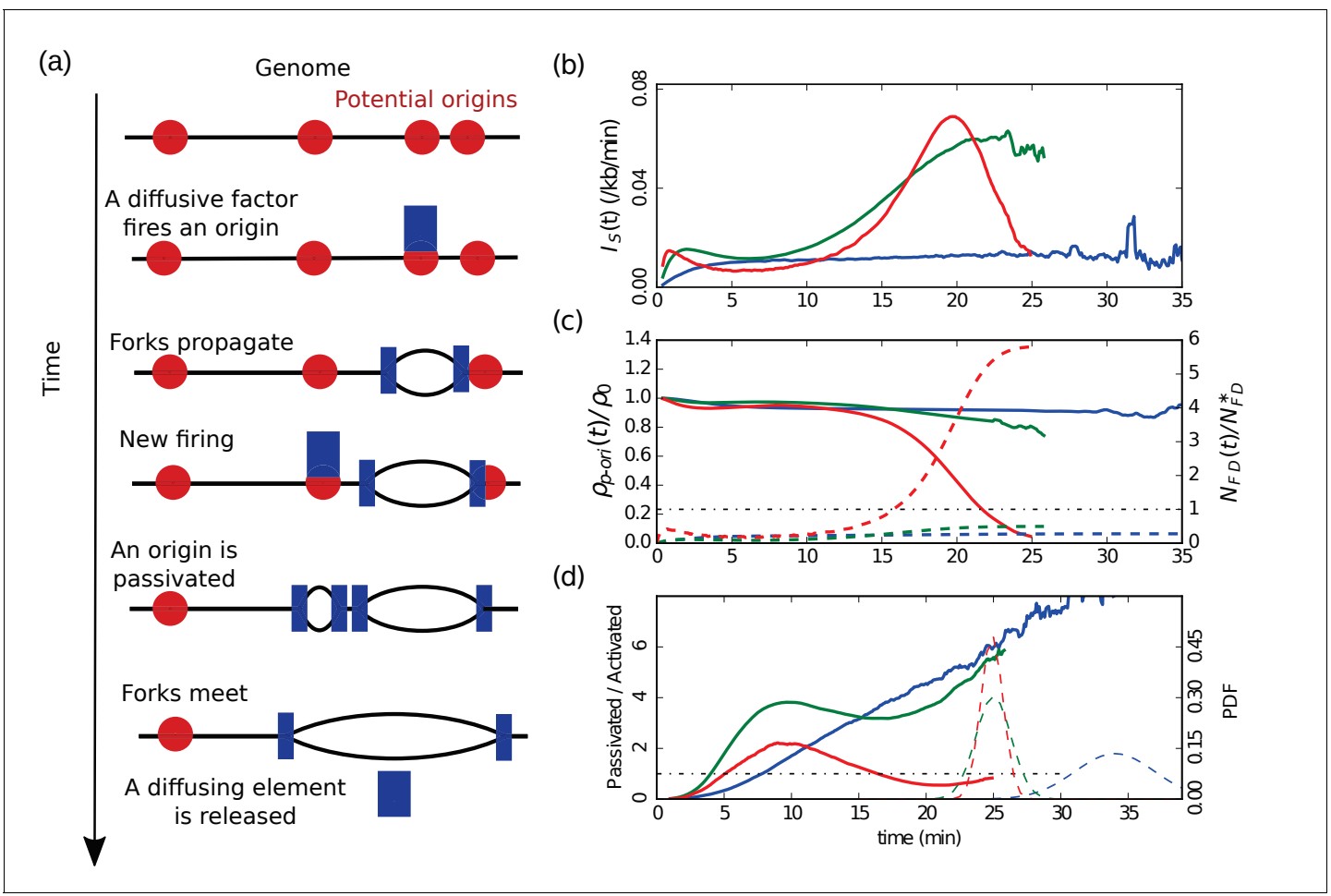

**Figure 1.** Emergence of a bell-shaped $I(t)$. (a) Sketch of the different steps of our modeling of replication initiation and propagation. (b) $I_S(t)$ (*Equation 1*) obtained from numerical simulations (Materials and methods) of one chromosome of length 3000 kb, with a fork speed $v = 0.6$ kb/min. The firing factors are loaded with a characteristic time of 3 min. From blue to green to red the interaction is increased and the number of firing factors is decreased: blue ($k_{on} = 5 \times 10^{-5}$ min$^{-1}$, $N_D^T = 1000$, $\rho_0 = 0.3$ kb$^{-1}$), green ($k_{on} = 6 \times 10^{-4}$ min$^{-1}$, $N_D^T = 250$, $\rho_0 = 0.5$ kb$^{-1}$), red ($k_{on} = 6 \times 10^{-3}$ min$^{-1}$, $N_D^T = 165$, $\rho_0 = 0.28$ kb$^{-1}$). (c) Corresponding normalized densities of *p-oris* (solid lines), and corresponding normalized numbers of free diffusing firing factors (dashed line): blue ($N_{FD}^* = 3360$), green ($N_{FD}^* = 280$), red ($N_{FD}^* = 28$); the horizontal dotted-dashed line corresponds to the critical threshold value $N_{FD}(t) = N_{FD}^*$. (d) Corresponding number of passivated origins over the number of activated origins (solid lines). Corresponding probability distribution functions (PDF) of replication time (dashed lines).

DOI: https://doi.org/10.7554/eLife.35192.003

where $N_{fired}(t, t+dt)$ is the number of *p-oris* fired between times $t$ and $t+dt$, and $L_{unrepDNA}(t)$ is the length of unreplicated DNA a time $t$. Then we propagate the forks along the chromosome with a constant speed $v$, and if two forks meet, the two half firing complexes are released and rapidly reform an active firing factor. Finally, we simulate the chromosomes as 1D chains where prior to entering S-phase, the *p-oris* are precisely localized. For *Xenopus* embryo, the *p-ori* positions are randomly sampled, so that each simulated S-phase corresponds to a different positioning of the *p-oris*. We compare results obtained with periodic or uniform *p-ori* distributions (Materials and methods). For *S. cerevisiae*, the *p-ori* positions, identical for each simulation, are taken from the OriDB database (*Siow et al., 2012*). As previously simulated in human (*Löb et al., 2016*), we model the entry in S-phase using an exponentially relaxed loading of the firing factors with a time scale shorter than the S-phase duration $T_{phase}$ (3 min for *Xenopus* embryo, where $T_{phase} \sim 30$ min, and 10 min for *S. cerevisiae*, where $T_{phase} \sim 60$ mins). After the short loading time, the total number of firing factors $N_D^T$ is constant. As shown in *Figure 1b* (see also *Figure 2*), the universal bell shape of the $I(t)$ curves (*Goldar et al., 2009*) spontaneously emerges from our model when going from weak to strong interaction, and decreasing the number of firing factors below the number of *p-oris*. The details of the firing factor loading dynamics do not affect the emergence of a bell shaped $I(t)$, even though it can modulate its precise shape, especially early in S-phase.

In a simple bimolecular context, the rate of origin firing is $i(t) = k_{on}N_{\text{p-ori}}(t)N_{FD}(t)$. The firing rate by element of unreplicated DNA is then given by

$$I(t) = k_{on}N_{FD}(t)\rho_{\text{p-ori}}(t), \tag{2}$$

where $\rho_{\text{p-ori}}(t) = N_{\text{p-ori}}(t)/L_{unrepDNA}(t)$. In the case of a strong interaction and a limited number of firing factors, all the diffusing factors react rapidly after loading and $N_{FD}(t)$ is small (*Figure 1 (c)*, dashed curves). Then follows a stationary phase where as long as the number of *p-oris* is high (*Figure 1 (c)*, solid curves), once a diffusing factor is released by the encounter of two forks, it reacts rapidly, and so $N_{FD}(t)$ stays small. Then, when the rate of fork mergers increases due to the fact that there are as many active forks but a smaller length of unreplicated DNA, the number of free firing factors increases up to $N_D^T$ at the end of S-phase. As a consequence, the contribution of $N_{FD}(t)$ to $I(t)$ in *Equation (2)* can only account for a monotonous increase during the S phase. For $I(t)$ to reach a maximum $I_{max}$ before the end of S-phase, we thus need that $\rho_{\text{p-ori}}(t)$ decreases in the late S-phase. This happens if the time to fire a *p-ori* is shorter than the time to replicate a typical distance between two neighboring *p-oris*. The characteristic time to fire a *p-ori* is $t_c = 1/k_{on}N_{FD}(t)$. The mean time for a fork to replicate DNA between two neighboring *p-oris* is $t_r = d(t)/v$, where $d(t)$ is the mean distance between unreplicated *p-oris* at time $t$. So the density of origins is constant as long as:

$$\frac{d(t)}{v} < \frac{1}{k_{on}N_{FD}(t)}, \tag{3}$$

or

$$N_{FD}(t) < N_{FD}^* = \frac{v}{k_{on}d(t)}. \tag{4}$$

Thus, at the beginning of the S-phase, $N_{FD}(t)$ is small, $\rho_{\text{p-ori}}(t)$ is constant (*Figure 1 (c)*, solid curves) and so $I_S(t)$ stays small. When $N_{FD}(t)$ starts increasing, as long as *Equation (4)* stays valid, $I_S(t)$ keeps increasing. When $N_{FD}(t)$ becomes too large and exceeds $N_{FD}^*$, then *Equation (4)* is violated and the number of *p-oris* decreases at a higher rate than the length of unreplicated DNA, and $\rho_{\text{p-ori}}(t)$ decreases and goes to zero (*Figure 1 (c)*, red solid curve). As $N_{FD}(t)$ tends to $N_D^T$, $I_S(t)$ goes to zero, and its global behavior is a bell shape (*Figure 1 (b)*, red). Let us note that if we decrease the interaction strength ($k_{on}$), then the critical $N_{FD}^*$ will increase beyond $N_D^T$ (*Figure 1 (c)*, dashed blue and green curves). $I_S(t)$ then monotonously increase to reach a plateau (*Figure 1 (b)*, green), or if we decrease further $k_{on}$, $I_S(t)$ present a very slow increasing behavior during the S-phase (*Figure 1 (b)*, blue). Now if we come back to strong interactions and increase the number of firing factors, almost all the *p-oris* are fired immediately and $I_S(t)$ drops to zero after firing the last *p-ori*.

Another way to look at the density of *p-oris* is to compute the ratio of the number of passivated origins by the number of activated origins (*Figure 1 (d)*). After the initial loading of firing factors,

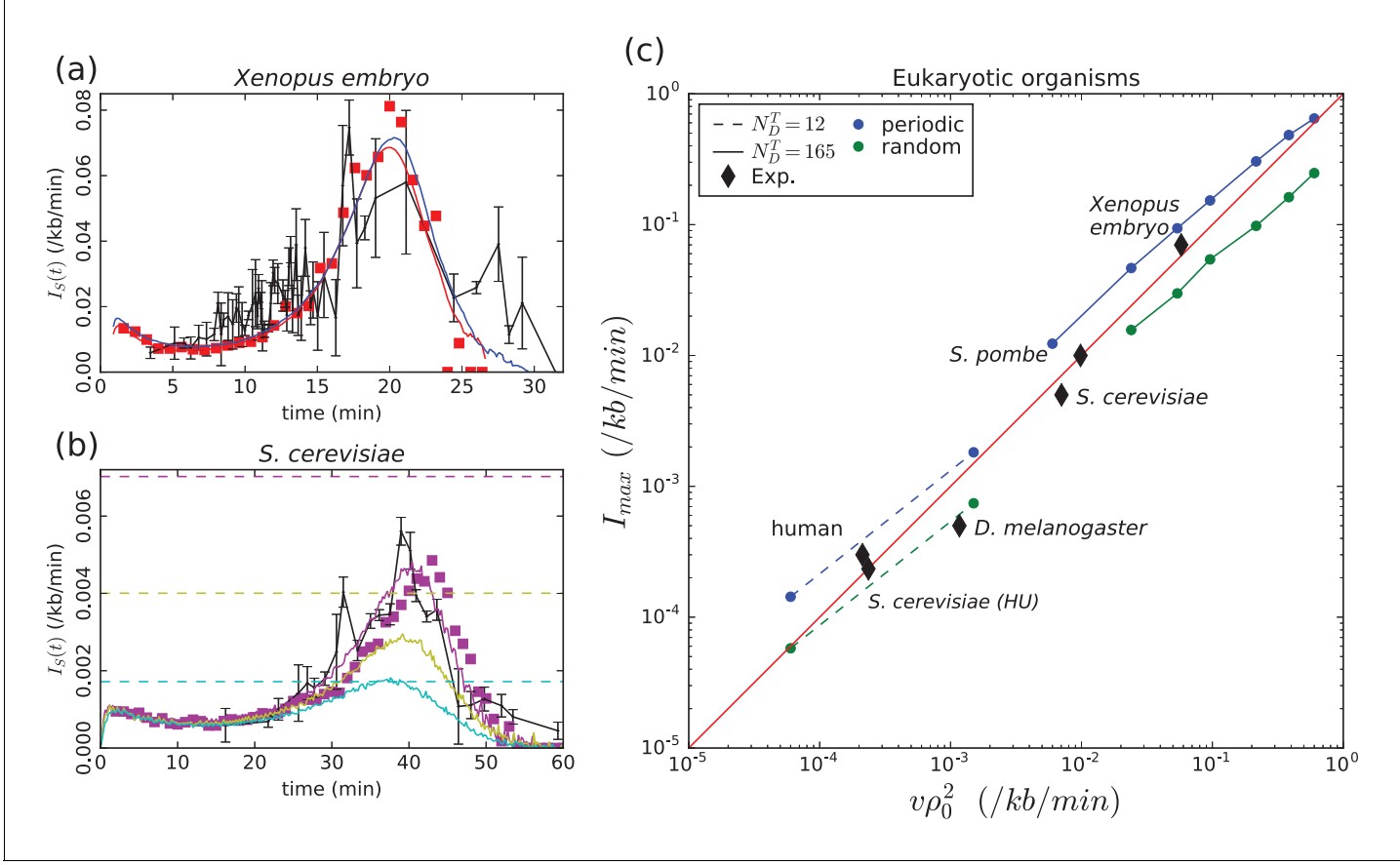

**Figure 2.** Model validation by experimental data. (a) *Xenopus* embryo: Simulated $I_S(t)$ (*Equation (1)*, Materials and methods) for a chromosome of length $L = 3000$ kb and a uniform distribution of *p-oris* (blue: $v = 0.6$ kb/min, $k_{on} = 3. \times 10^{-3}$ min$^{-1}$, $N_D^T = 187$, $\rho_0 = 0.70$ kb$^{-1}$) or a periodic distribution of *p-oris* (red: $v = 0.6$ kb/min, $k_{on} = 6 \times 10^{-3}$ min$^{-1}$, $N_D^T = 165$, $\rho_0 = 0.28$ kb$^{-1}$); (red squares) 3D simulations with the same parameter values as for periodic *p-ori* distribution; (black) experimental $I(t)$: raw data obtained from *Goldar et al. (2009)* were binned in groups of 4 data points; the mean value and standard error of the mean of each bin were represented. (b) *S. cerevisiae*: Simulated $I_S(t)$ (Materials and methods) for the 16 chromosomes with the following parameter values: $v = 1.5$ kb/min, $N_D^T = 143$, $k_{on} = 3.6 \times 10^{-3}$ min$^{-1}$, when considering only Confirmed origins (light blue), Confirmed and Likely origins (yellow) and Confirmed, Likely and Dubious origins (purple); the horizontal dashed lines mark the corresponding predictions for $I_{max}$ (*Equation 5*); (purple squares) 3D simulations with the same parameter values considering Confirmed, Likely and Dubious origins; (black) experimental $I(t)$ from *Goldar et al. (2009)*. (c) Eukaryotic organisms: $I_{max}$ as a function of $v\rho_0^2$; (squares and bullets) simulations performed for regularly spaced origins (blue) and uniformly distributed origins (green) (Materials and methods) with two sets of parameter values: $L = 3000$ kb, $v = 0.6$ kb/min, $k_{on} = 1.2 \times 10^{-2}$ min$^{-1}$ and $N_D^T = 12$ (dashed line) or 165 (solid line); (black diamonds) experimental data points for *Xenopus* embryo, *S. cerevisiae*, *S. cerevisae* grown in Hydroxyurea (HU), *S. pombe*, *D. melanogaster*, human (see text and *Table 1*). The following figure supplement is available for *Figure 2*.
DOI: https://doi.org/10.7554/eLife.35192.004

The following source data and figure supplement are available for figure 2:

**Source data 1.** Data file for the experimental *Xenopus* $I(t)$ in *Figure 2 (a)*.
DOI: https://doi.org/10.7554/eLife.35192.006
**Source data 2.** Data file for the experimental *S.*
DOI: https://doi.org/10.7554/eLife.35192.007
**Source data 3.** Data file for the experimental parameters used in *Figure 2 (c)*.
DOI: https://doi.org/10.7554/eLife.35192.008
**Figure supplement 1.** Different steps of the interaction between diffusing elements and origins of replication.
DOI: https://doi.org/10.7554/eLife.35192.005

this ratio is higher than one. For weak and moderate interactions (**Figure 1 (d)**, blue and green solid curves, respectively), this ratio stays bigger than one during all the S-phase, where $I_S(t)$ was shown to be monotonously increasing (**Figure 1 (b)**). For a strong interaction (**Figure 1 (b)**, red solid curve), this ratio reaches a maximum and then decreases below one, at a time corresponding to the maximum observed in $I_S(t)$ (**Figure 1 (d)**, red solid curve). Hence, the maximum of $I(t)$ corresponds to a switch of the balance between origin passivation and activation, the latter becoming predominant in late S-phase. We have seen that up to this maximum $\rho_{\text{p-ori}}(t) \approx cte \approx \rho_0$, so $I_S(t) \approx k_{on}\rho_0 N_F(t)$. When $N_{FD}(t)$ reaches $N_{FD}^\star$, then $I_S(t)$ reaches its maximum value:

$$I_{max} = k_{on}\rho_0 N_{FD}^\star \approx \frac{\rho_0 v}{d(t)} \approx v\rho_0^2, \tag{5}$$

where we have used the approximation $d(t) \approx d(0) = 1/\rho_0$ (which is exact for periodically distributed *p-oris*). $I_{max}$ can thus be predicted from two measurable parameters, providing a direct test of the model.

## Comparison with different eukaryotes

### *Xenopus* embryo

Given the huge size of *Xenopus* embryo chromosomes, to make the simulations more easily tractable, we rescaled the size $L$ of the chromosomes, $k_{on}$ and $N_D^T$ to keep the duration of S-phase $T_{phase} \approx L/2vN_D^T$ and $I(t)$ (**Equation (2)**) unchanged ($L \to \alpha L$, $N_D^T \to \alpha N_D^T$, $k_{on} \to k_{on}/\alpha$). In **Figure 2 (a)** are reported the results of our simulations for a chromosome length $L = 3000$ kb. We see that a good agreement is obtained with experimental data (**Goldar et al., 2009**) when using either a uniform distribution of *p-oris* with a density $\rho_0 = 0.70$ kb$^{-1}$ and a number of firing factors $N_D^T = 187$, or a periodic distribution with $\rho_0 = 0.28$ kb$^{-1}$ and $N_D^T = 165$. A higher density of *p-oris* was needed for uniformly distributed *p-oris* where $d(t)$ (slightly) increases with time, than for periodically distributed *p-oris* where $d(t)$ fluctuates around a constant value $1/\rho_0$. The uniform distribution, which is the most natural to simulate *Xenopus* embryo replication, gives a density of activated origins of 0.17 kb$^{-1}$ in good agreement with DNA combing data analysis (**Herrick et al., 2002**) but twice lower than estimated from real time replication imaging of surface-immobilized DNA in a soluble *Xenopus* egg extract system (**Loveland et al., 2012**). Note that in the latter work, origin licensing was performed in condition of incomplete chromatinization and replication initiation was obtained using a nucleoplasmic extract system with strong initiation activity, which may explain the higher density of activated origins observed in this work.

### S. cerevisiae

To test the robustness of our minimal model with respect to the distribution of *p-oris*, we simulated the replication in *S. cerevisiae*, whose *p-oris* are known to be well positioned as reported in OriDB (**Siow et al., 2012**). 829 *p-oris* were experimentally identified and classified into three categories: Confirmed origins (410), Likely origins (216), and Dubious origins (203). When comparing the results obtained with our model to the experimental $I(t)$ data (**Goldar et al., 2009**) (**Figure 2 (b)**), we see that to obtain a good agreement we need to consider not only the Confirmed origins but also the Likely and the Dubious origins. This shows that in the context of our model, the number of *p-oris* required to reproduce the experimental $I(t)$ curve in *S. cerevisiae* exceeds the number of Confirmed and Likely origins. Apart from the unexpected activity of Dubious origins, the requirement for a larger number of origins can be met by some level of random initiation (**Czajkowsky et al., 2008**) or initiation events away from mapped origins due to helicase mobility (**Gros et al., 2015**; **Hyrien, 2016a**). If fork progression can push helicases along chromosomes instead of simply passivating them, there will be initiation events just ahead of progressing forks. Such events are not detectable by the replication profiling experiments used to determine $I(t)$ in **Figure 2(b)** and thus not accounted for by $I_{max}$. Given the uncertainty in replication fork velocity (a higher fork speed would require only Confirmed and Likely origins) and the possible experimental contribution of the *p-oris* in the rDNA part of chromosome 12 (not taken into account in our modeling), this conclusion needs to be confirmed in future experiments. It is to be noted that even if 829 *p-oris* are needed, on average only 352 origins have fired by the end of S-phase. For *S. cerevisiae* with well positioned *p-oris*, we have checked the robustness of our results with respect to a stochastic number of firing

factors $N_D^T$ from cell to cell (Poisson distribution, *Iyer-Biswas et al. (2009)*). We confirmed the $I(t)$ bell shape with a robust duration of the S-phase of $58.6 \pm 4.3$ min as compared to $58.5 \pm 3.3$ min obtained previously with a constant number of firing factors. Interestingly, in an experiment where hydroxyurea (HU) was added to the yeast growth media, the sequence of activation of replication origins was shown to be conserved even though $T_{phase}$ was lengthened from 1 hr to 16 hr (*Alvino et al., 2007*). HU slows down the DNA synthesis to a rate of $\sim 50 \, \text{bp} \, \text{min}^{-1}$ corresponding to a 30-fold decrease of the fork speed (*Sogo et al., 2002*). Up to a rescaling of time, the replication kinetics of our model is governed by the ratio between replication fork speed and the productive-interaction rate $k_{on}$ (neglecting here the possible contribution of the activation dynamics of firing factors). Hence, our model can capture the observation of *Alvino et al. (2007)* when considering a concomitant fork slowing down and $k_{on}$ reduction in response to HU, which is consistent with the molecular action of the replication checkpoint induced by HU (*Zegerman and Diffley, 2010*). In a model where the increase of $I(t)$ results from the import of replication factors, the import rate would need to be reduced by the presence of HU in proportion with the lengthening of S-phase in order to maintain the pattern of origin activations. Extracting $I(t)$ from experimental replication data for cells grown in absence (HU$^-$) or presence (HU$^+$) (*Alvino et al., 2007*), we estimated $I_{max}^{\text{HU}-} \sim 6.0 \, \text{Mb}^{-1} \, \text{min}^{-1}$ and $I_{max}^{\text{HU}+} \sim 0.24 \, \text{Mb}^{-1} \, \text{min}^{-1}$ for HU$^-$ and HU$^+$ cells, respectively. The ratio $I_{max}^{\text{HU}-}/I_{max}^{\text{HU}+} \simeq 24.8 \sim v^{\text{HU}-}/v^{\text{HU}+}$ is quite consistent with the prediction of the scaling law (*Equation (5)*) for a constant density of *p-oris*.

## *D. melanogaster* and human

We gathered from the literature experimental estimates of $I_{max}$, $\rho_0$ and $v$ for different eukaryotic organisms (*Table 1*). As shown in *Figure 2 (c)*, when plotting $I_{max}$ vs $v\rho_0^2$, all the experimental data points remarkably follow the diagonal trend indicating the validity of the scaling law (*Eq. (5)*) for all considered eukaryotes. We performed two series of simulations for fixed values of parameters $k_o$, $N_D^T$ and $v$ and decreasing values of $\rho_0$ with both periodic distribution (blue) and uniform (green) distributions of *p-oris* (*Figure 2 (c)*). The first set of parameters was chosen to cover high $I_{max}$ values similar the one observed for *Xenopus* embryo (bullets, solid lines). When decreasing $\rho_0$, the number of firing factors becomes too large and $I(t)$ does no longer present a maximum. We thus decreased the value of $N_D^T$ keeping all other parameters constant (boxes, dashed line) to explore smaller values of $I_{max}$ in the range of those observed for human and *D. melanogaster*. We can observe that experimental data points' deviation from *Equation (5)* is smaller than the deviation due to specific *p-oris* distributions.

**Table 1.** Experimental data for various eukaryotic organisms with genome length $L$ (*Mb*), replication fork velocity $v$ (kb/min), number of *p-oris* ($N_{\text{p-ori}}(t=0)$), $\rho_0 = N_{\text{p-ori}}(t=0)/L$ (kb$^{-1}$) and $I_{max}$ (Mb$^{-1}$min$^{-1}$).

All $I_{max}$ data are from *Goldar et al. (2009)*, except for *S. cerevisiae* grown in presence or absence of hydroxyurea (HU) which were computed from the replication profile of *Alvino et al. (2007)*. For *S. cerevisiae* and *S. pombe*, Confirmed, Likely, and Dubious origins were taken into account. For *D. melanogaster*, $N_{\text{p-ori}}(t=0)$ was obtained from the same Kc cell type as the one used to estimate $I_{max}$. For *Xenopus* embryo, we assumed that a *p-ori* corresponds to a dimer of MCM2-7 hexamer so that $N_{\text{p-ori}}(t=0)$ was estimated as a half of the experimental density of MCM3 molecules reported for *Xenopus* sperm nuclei DNA in *Xenopus* egg extract (*Mahbubani et al., 1997*). For human, we averaged the number of origins experimentally identified in K562 (62971) and in MCF7 (94195) cell lines.

| | $L$ | $v$ | $N_{\text{p-ori}}$ | $\rho_0$ | $I_{max}$ | Ref. |
|---|---|---|---|---|---|---|
| *S. cerevisiae* | 12.5 | 1.60 | 829 | 0.066 | 6.0 | *Sekedat et al. (2010)* and *Siow et al. (2012)* |
| *S. cerevisiae* in presence of HU | 12.5 | 0.05 | 829 | 0.066 | 0.24 | *Alvino et al. (2007)*. Same $N_{\text{p-ori}}$ and $\rho_0$ as *S. cerevisiae* in normal growth condition. |
| *S. pombe* | 12.5 | 2.80 | 741 | 0.059 | 10.0 | *Siow et al. (2012)* and *Kaykov and Nurse (2015)* |
| *D. melanogaster* | 143.6 | 0.63 | 6184 | 0.043 | 0.5 | *Ananiev et al. (1977)* and *Cayrou et al. (2011)* |
| human | 6469.0 | 1.46 | 78000 | 0.012 | 0.3 | *Conti et al. (2007)* and *Martin et al. (2011)* |
| *Xenopus* sperm | 2233.0 | 0.52 | 744333 | 0.333 | 70.0 | *Mahbubani et al. (1997)* and *Loveland et al. (2012)* |

DOI: https://doi.org/10.7554/eLife.35192.009

Note that in human it was suggested that early and late replicating domains could be modeled by spatial inhomogeneity of the *p-ori* distribution along chromosomes, with a high density in early replicating domains ($\rho_{0,early} = 2.6$ ORC/100 kb) and a low density in late replicating domains ($\rho_{0,late} = 0.2$ ORC/100 kb) (*Miotto et al., 2016*). If low- and high-density regions each cover one half of the genome and $\rho_{0,early} \gg \rho_{0,late}$, most *p-oris* are located in the high-density regions and the origin firing kinetics ($N_{fired}(t, t + dt)$) will mainly come from initiation in these regions. However, the length of unreplicated DNA also encompasses the late replicating domains resulting in a lowering of the global $I(t)$ by at least a factor of 2 (*Equation (1)*). Hence, in the context of our model $I_{max} \lesssim 0.5 v \rho_{early}^2$. Interestingly, considering the experimental values for the human genome ($I_{max} = 0.3 \mathrm{Mb}^{-1}\mathrm{min}^{-1}$ and $v = 1.46 \mathrm{kb\,min}^{-1}$, *Table 1*), this leads to $\rho_{0,early} \gtrsim 2.3$ Ori/100 kb, in good agreement with the esti- mated density of 2.6 ORC/100 kb (*Miotto et al., 2016*). Inhomogeneities in origin density could cre- ate inhomogeneities in firing factor concentration that would further enhance the replication kinetics in high density regions, possibly corresponding to early replication foci.

## Discussion

To summarize, we have shown that within the framework of 1D nucleation and growth models of DNA replication kinetics (*Herrick et al., 2002*; *Jun and Bechhoefer, 2005*), the sufficient conditions to obtain a universal bell shaped $I(t)$ as observed in eukaryotes are a strong bimolecular reaction between localized *p-oris* and limiting origin firing factors that travel with replication forks and are released at termination. Under these conditions, the density of *p-oris* naturally decreases by the end of the S-phase and so does $I_S(t)$. Previous models in *Xenopus* embryo (*Goldar et al., 2008*; *Gauthier and Bechhoefer, 2009*) assumed that all sites contained a *p-ori* implying that the time $t_r$ to replicate DNA between two neighboring *p-oris* was close to zero. This clarifies why they needed some additional mechanisms to explain the final decrease of the firing rate. Moreover, our model predicts that the maximum value for $I(t)$ is intimately related to the density of *p-oris* and the fork speed (*Equation (5)*), and we have shown that without free parameter, this relationship holds for five species with up to a 300-fold difference of $I_{max}$ and $v\rho_0^2$ (*Table 1*, *Figure 2 (c)*).

Our model assumes that all *p-oris* are governed by the same rule of initiation resulting from phys- icochemically realistic particulars of their interaction with limiting replication firing factors. Any spa- tial inhomogeneity in the firing rate $I(x, t)$ along the genomic coordinate in our simulations thus reflects the inhomogeneity in the distribution of the potential origins in the genome. In yeast, repli- cation kinetics along chromosomes were robustly reproduced in simulations where each replication origin fires following a stochastic law with parameters that change from origin to origin (*Yang et al., 2010*). Interestingly, this heterogeneity between origins is captured by the Multiple-Initiator Model where origin firing time distribution is modeled by the number of MCM2-7 complexes bound at the origin (*Yang et al., 2010*; *Das et al., 2015*). In human, early and late replicating domains could be modeled by the spatial heterogeneity of the origin recognition complex (ORC) distribution (*Miotto et al., 2016*). In these models, MCM2-7 and ORC have the same status as our *p-oris*, they are potential origins with identical firing properties. Our results show that the universal bell-shaped temporal rate of replication origin firing can be explained irrespective of species-specific spatial het- erogeneity in origin strength. Note, however, that current successful modeling of the chromosome organization of DNA replication timing relies on heterogeneities in origins' strength and spatial dis- tributions (*Bechhoefer and Rhind, 2012*).

To confirm the simple physical basis of our modeling, we used molecular dynamics rules as previ- ously developed for *S. cerevisiae* (*Arbona et al., 2017*) to simulate S-phase dynamics of chromo- somes confined in a spherical nucleus. We added firing factors that are free to diffuse in the covolume left by the chain and that can bind to proximal *p-oris* to initiate replication, move along the chromosomes with the replication forks and be released when two fork merges. As shown in *Fig- ure 2 (a,b)* for *Xenopus* embryo and *S. cerevisiae*, results confirmed the physical relevance of our minimal modeling and the validity of its predictions when the 3D diffusion of the firing factors is explicitly taken into account. Modeling of replication timing profiles in human was recently success- fully achieved in a model with both inhibition of origin firing 55 kb around active forks, and an enhanced firing rate further away up to a few 100 kb (*Löb et al., 2016*) as well as in models that do not consider any inhibition nor enhanced firing rate due to fork progression (*Gindin et al., 2014*;

*Miotto et al., 2016*). These works illustrate that untangling spatio-temporal correlations in replication kinetics is challenging. 3D modeling opens new perspectives for understanding the contribution of firing factor transport to the correlations between firing events along chromosomes. For example in *S. cerevisiae* (*Knott et al., 2012*) and in *S. pombe* (*Kaykov and Nurse, 2015*), a higher firing rate has been reported near origins that have just fired (but see *Yang et al. (2010)*). In mammals, megabase chromosomal regions of synchronous firing were first observed a long time ago (*Huberman and Riggs, 1968*; *Hyrien, 2016b*) and the projection of the replication program on 3D models of chromosome architecture was shown to reproduce the observed S-phase dynamics of replication foci (*Löb et al., 2016*). Recently, profiling of replication fork directionality obtained by Okazaki fragment sequencing have suggested that early firing origins located at the border of Topologically Associating Domains (TADs) trigger a cascade of secondary initiation events propagating through the TAD (*Petryk et al., 2016*). Early and late replicating domains were associated with nuclear compartments of open and closed chromatin (*Ryba et al., 2010*; *Boulos et al., 2015*; *Goldar et al., 2016*; *Hyrien, 2016b*). In human, replication timing U-domains (0.1–3 Mb) were shown to correlate with chromosome structural domains (*Baker et al., 2012*; *Moindrot et al., 2012*; *Pope et al., 2014*) and chromatin loops (*Boulos et al., 2013*, *Boulos et al., 2014*).

Understanding to which extent spatio-temporal correlations of the replication program can be explained by the diffusion of firing factors in the tertiary chromatin structure specific to each eukaryotic organism is a challenging issue for future work.

## Materials and methods

### Well-mixed model simulations

Each model simulation allows the reconstruction of the full replication kinetics during one S-phase. Chromosome initial replication state is described by the distribution of *p-oris* along each chromosomes. For *Xenopus* embryo, *p-ori* positions are randomly determined at the beginning of each simulation following two possible scenarios:

- For the uniform distribution scenario, $L\rho_0$ origins are randomly positions in the segment $[0, L]$, where $\rho_0$ is the average density of potential origins and $L$ the total length of DNA.
- For the periodic distribution scenario, exactly one origin is positioned in every non-overlapping $1/\rho_0$ long segment. Within each segment, the position of the origin is chosen randomly in order to avoid spurious synchronization effects.

For yeast, the *p-ori* positions are identical in each S-phase simulations and correspond to experimentally determined positions reported in OriDB (*Siow et al., 2012*). The simulation starts with a fixed number $N_D^T$ of firing factors that are progressively made available as described in Results. At every time step $t = ndt$, each free firing factor (available factors not bound to an active replication fork) has a probability to fire one of the $N_{p-ori}(t)$ *p-oris* at unreplicated loci given by:

$$1 - (1 - k_{on}dt)^{N_{p-ori}(t)}. \tag{6}$$

A random number is generated, and if it is inferior to this probability, an unreplicated *p-ori* is chosen at random, two diverging forks are created at this locus and the number of free firing factors decreases by 1. Finally, every fork is propagated by a length $vdt$ resulting in an increase amount of DNA marked as replicated and possibly to the passivation of some *p-oris*. If two forks meet they are removed and the number of free firing factors increases by 1. Forks that reach the end of a chromosome are discarded. The numbers of firing events ($N_{fired}(t)$), origin passivations, free firing factors ($N_{FD}(t)$) and unreplicated *p-oris* ($N_{p-ori}(t)$) as well as the length of unreplicated DNA ($L_{unrepDNA}(t)$) are recorded allowing the computation of $I_S(t)$ (*Eq. (1)*), the normalized density of *p-oris* ($\rho_{p-ori}(t))/\rho_0$), the normalized number of free firing factors ($N_{FD}(t)/N_{FD}^*(t)$) and the ratio between the number of origin passivations and activations. Simulation ends when all DNA has been replicated, which define the replication time.

### 3D model simulations

Replication kinetics simulation for the 3D model follows the same steps as in the well-mixed model except that the probability that a free firing factor activates an unreplicated *p-ori* depends on their

distance $d$ obtained from a molecular dynamic simulation performed in parallel to the replication kinetics simulation. We used HOOMD-blue (*Anderson et al., 2008*) with parameters similar to the ones previously considered in *Arbona et al. (2017)* to simulate chromosome conformation dynamics and free firing factor diffusion within a spherical nucleus of volume $V_N$. The details of the interaction between the diffusing firing factors and the *p-oris* is illustrated in *Figure 2—figure supplement 1*. Given a capture radius $r_c$ set to two coarse grained chromatin monomer radiuses, when a free firing factor is within the capture volume $V_c = \frac{4}{3}\pi r_c^3$ around an unreplicated *p-ori* ($d<r_c$), it can activate the origin with a probability $p$. In order to have a similar firing activity as in the well-mixed model, $r_c$ and $p$ were chosen so that $pV_c/V_N$ takes a value comparable to the $k_{on}$ values used in the well-mixed simulations.

For each set of parameters of the well-mixed and 3D models, we reported the mean curves obtained over a number of independent simulations large enough so that the noisy fluctuations of the mean $I_S(t)$ are small compared to the average bell-shaped curve. The complete set of parameters for each simulation series is provided in *Supplementary file 1*. The scripts used to extract yeast $I(t)$ from the experimental data of *Alvino et al. (2007)* can be found here https://github.com/ jeam-mimi/ifromprof/blob/master/notebooks/exploratory/Alvino_WT.ipynb (yeast in normal growth conditions) and here https://github.com/jeammimi/ifromprof/blob/master/notebooks/exploratory/Alvino_H.ipynb (yeast grown grown in Hydroxyurea) (*Arbona and Goldar, 2018*). A copy is archived at https://github.com/elifesciences-publications/ifromprof.

## Acknowledgements

We thank F. Argoul for helpful discussions. This work was supported by Institut National du Cancer (PLBIO16-302), Fondation pour la Recherche Médicale (DEI20151234404) and Agence Nationale de la Recherche (ANR-15-CE12-0011-01). BA acknowledges support from Science and Technology Commission of Shanghai Municipality (15520711500) and Joint Research Institute for Science and Society (JoRISS). We gratefully acknowledge support from the PSMN (Pôle Scientifique de Modélisation Numérique) of the ENS de Lyon for the computing resources. We thank BioSyL Federation and Ecofect LabEx (ANR-11-LABX-0048) for inspiring scientific events.

## Additional information

### Funding

| Funder | Grant reference number | Author |
|---|---|---|
| Institut National Du Cancer | PLBIO16-302 | Olivier Hyrien<br>Benjamin Audit |
| Fondation pour la Recherche Médicale | DEI20151234404 | Arach Goldar<br>Olivier Hyrien<br>Benjamin Audit |
| Agence Nationale de la Recherche | ANR-15-CE12-0011-01 | Olivier Hyrien<br>Alain Arneodo<br>Benjamin Audit |
| Joint Research Institute for Science and Society | JoRISS 2017-2018 | Benjamin Audit |

The funders had no role in study design, data collection and interpretation, or the decision to submit the work for publication.

### Author contributions

Jean-Michel Arbona, Data curation, Software, Formal analysis, Investigation, Visualization, Methodology, Writing—original draft, Writing—review and editing; Arach Goldar, Conceptualization, Resources, Data curation, Validation, Methodology; Olivier Hyrien, Benjamin Audit, Conceptualization, Supervision, Funding acquisition, Validation, Writing—original draft, Writing—review and editing; Alain Arneodo, Conceptualization, Supervision, Writing—original draft, Writing—review and editing

**Author ORCIDs**
Jean-Michel Arbona (iD) http://orcid.org/0000-0001-6166-9056
Olivier Hyrien (iD) http://orcid.org/0000-0001-8879-675X
Benjamin Audit (iD) http://orcid.org/0000-0003-2683-9990

**Decision letter and Author response**
Decision letter https://doi.org/10.7554/eLife.35192.013
Author response https://doi.org/10.7554/eLife.35192.014

## Additional files

### Supplementary files

• Supplementary file 1. This file provides: the parameter values used for all the simulations in *Figures 1* and *2*; the list of all the symbols used in the main text and their meanings.
DOI: https://doi.org/10.7554/eLife.35192.010

• Transparent reporting form
DOI: https://doi.org/10.7554/eLife.35192.011

### Data availability

All experimental data analyzed in this study are included in the manuscript. Source data files have been provided for Figure 2.

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
