## [Decision Letter]

Thank you for submitting your article "The eukaryotic bell-shaped temporal rate of DNA replication origin firing emanates from a balance between origin activation and passivation" for consideration by *eLife*. Your article has been reviewed by three peer reviewers, and the evaluation has been overseen by a Reviewing Editor and Kevin Struhl as the Senior Editor. The reviewers have opted to remain anonymous.

The reviewers have discussed the reviews with one another and the Reviewing Editor has drafted this decision to help you prepare a revised submission.

Based on the three positive reviews, it is possible that this model can by published in *eLife*, but there are some issues that two of the reviewers raise that require a response. Once a response is received, the paper will be re-considered.

Summary:

This appealing paper advances a new hypothesis to explain the observed, apparently quite general phenomenon in eukaryotic replication that the initiation rate of origin firing (relative to the amount of unreplicated DNA) decreases at the end of S-phase after having increased substantially throughout the first part of S-phase. There is agreement on the mechanism of the increase, but three different groups (one of which includes two of the present authors) have advanced three different hypotheses (subdiffusive motion, a dependence on replication fork density of firing factor affinity for *p-oris*, or inhomogeneous firing probabilities). The present paper proposes a fourth, that the limiting factor is the finite average spacing between potential origins (*p-oris*) and makes a case that this new hypothesis is both simple and natural. The evidence presented is a mix of heuristic argument, simulation, and a limited comparison of experimental data. The major experimental test is of a simple relation derived by the authors, *I_max_ ~ v ρ_0_^2^*, where *I_max_* is the maximum initiation rate, *v* the fork velocity, and *ρ_0_* the density of potential origins at the start of S-phase. The authors further look at a 3d simulation of the diffusion process in a simple model where all origins are treated on an equal footing and find a qualitative agreement in the *I(t)* curves.

The paper thus gives a simple model that advances our understanding of the replication process, adding a reasonable dynamical model to explain kinetics, and providing at least some experimental support-probably not enough to be completely convincing on its own but enough to make others take the hypothesis seriously and inspire further experimental tests. It is thus a nice advance.

Having said all of this, there are questions / reservations about some of the details as outlined below.

Essential revisions:

1) In the figures given for *Xenopus laevis* in Table 1, the value of *ρ_0_*is given as 0.333/kb, with Loveland et al. the reference. In that reference, though, Figure 3D shows only that the minimum average distance between fired origins decreases to 3kb. This implies only a lower bound on *ρ_0_*, since there may be passive replication in those experiments.

2) The 3D simulations, if they are understood correctly, will fail to reproduce the known genome-position dependence of firing times. Put another way, the authors argue in the Discussion section (second paragraph) that their modeling implies that all *p-oris* are the same. But in the *S. cerevisiae* data (and for other organisms), there are known dependences of median firing time on genome position. It may be that the model set forth here does a good job explaining the *I(t)* dependence but not the full *I(x,t)* dependence, where x is the genome position.

3) In a related point, the authors speculate that enhanced firing rates could result from diffusion of factors released. However, there is also evidence that chromatin looping can inhibit the firing of neighboring origins. Both effects could be present, suggesting that untangling spatiotemporal correlations might be subtle.

4) When the authors modeled replication in the presence of HU, it appears that the only change made in the parameters from unperturbed replication was the speed of replication forks. Is this correct? If so, it is surprising, as activation of late-firing origins are suppressed or delayed in HU, and according to Figure 1a, one might expect less origins to be passivated with slower replication forks in HU. The authors need to comment on this.

5) Figure 2B: It was unexpected that dubious origins needed to be included for better modeling. The authors need to discuss potential reasons for this.

6) It has been proposed that DNA replication takes place at replication foci in vivo, where replication factors are highly concentrated. Based on the authors' model that the localization of origins and recycling of replication factors can explain most of DNA replication kinetics, the authors need to discuss how the presence of replication foci would affect origin usage and replication kinetics.

7) The paper does not cite a published model for DNA replication timing by Miotto et al., 2016 that essentially states that there are more ORC sites than are utilized during S phase and early replicating regions at the beginning of S phase is favored simply because there are far more ORC sites, whereas firing from relatively few ORC sites in late replication regions is due to increased time and the unavailability of ORC sites previously replicated. This paper should be cited and discussed to compare it to the proposed model.

---

## [Author Response]

Essential revisions:1) In the figures given for Xenopus laevis in Table 1, the value of ρ_0_is given as 0.333/kb, with Loveland et al. the reference. In that reference, though, Figure 3D shows only that the minimum average distance between fired origins decreases to 3kb. This implies only a lower bound on ρ_0_, since there may be passive replication in those experiments.

We used as a proxy for the number of potential origins in *Xenopus* embryo the highest density (0.333 kb^−1^, Loveland et al., (2012)) of activated replication origins reported by the experimental studies of Loveland et al., (2012) and Herrick et al., (2002). We agree that this approach only provides a lower bound on ρ_0_ and we mentioned it in Table 1: “For *Xenopus* embryo, we used the experimental density of activated origins to estimate *N_p-ori_(t = 0)* which is probably lower than the true number of *p-oris.”* We had overlooked the work of Mahbubani et al., (1997) that estimated that, on average, one MCM3 molecule was bound to every 1.5 kbp of DNA from demembranated *Xenopus* sperm nuclei DNA in *Xenopus* egg extract. Assuming that a potential origin corresponds to a dimer of MCM2-7 hexamers, this experimental quantification provides an estimate of one *p-ori* every 3 kbp i.e. ρ_0_ = 0.333 kb^−1^. Note that in the work of Loveland et al., (2012), licensing of λ-phage DNA was performed in condition of incomplete chromatinization and replication initiation was obtained using a nucleoplasmic extract system with strong initiation activity, which may explain the high density of activated origins observed in this work.

We modified Table 1 caption accordingly. We replaced:

“For *Xenopus* embryo, we used the experimental density of activated origins to estimate *N_p-ori_(t=0)* which is probably lower than the true number of *p-oris.”*

with:

“For *Xenopus* embryo, we assumed that a *p-ori* corresponds to a dimer of MCM2-7 hexamer so that *N_p-ori_(t=0)* was estimated as a half of the experimental density of MCM3 molecules reported for *Xenopus* sperm nuclei DNA in *Xenopus* egg extract (Mahbubani et al., 1997).”

We added the following comment at the end the “*Xenopus* embryo” paragraph in the Results section:

“Note that in the latter work, origin licensing was performed in condition of incomplete chromatinization and replication initiation was obtained using a nucleoplasmic extract system with strong initiation activity, which may explain the higher density of activated origins observed in this work.”

2) The 3D simulations, if they are understood correctly, will fail to reproduce the known genome-position dependence of firing times. Put another way, the authors argue in the Discussion section (second paragraph) that their modeling implies that all p-oris are the same. But in the S. cerevisiae data (and for other organisms), there are known dependences of median firing time on genome position. It may be that the model set forth here does a good job explaining the I(t) dependence but not the full I(x,t) dependence, where x is the genome position.

The 1D simulations as well as the 3D simulations indeed do not consider any heterogeneity between potential origins properties. Hence, neglecting origin passivation, the median firing times for all the origins are identical. Any inhomogeneity in *I(x,t)* in our simulations thus reflects the inhomogeneity in the distribution of the potential origins in the genome, and not the heterogeneity of origin strengths. We are sorry if our presentation suggested that all yeast origins behave the same. In yeast, replication kinetics along chromosomes were robustly reproduced in simulations where each replication origin fires following a stochastic law with parameters that change from origin to origin (Yang et al., 2010). Interestingly, this heterogeneity between origins is captured by the Multiple-Initiator Model (MIM) where origin firing time distribution is modeled by the number of MCM2-7 complexes bound at the origin (Yang et al., 2010). In human, early and late replicating domains could be modeled by the spatial heterogeneity of the origin recognition complex (ORC) distribution (Miotto et al., 2016; see answer to point 7). In these models, MCM2-7 and ORC have the same status as our *p-oris,* they are potential origins with identical firing properties. Our results show that the universal bell-shaped temporal rate of replication origin firing can be explained irrespective of species-specific spatial heterogeneity in origin strength.

We have replaced:

“In contrast with models where replication kinetics is explained by properties specific to each *p-oris* (Bechhoeffer and Rhind, 2012), our model assumes that all *p-oris* are governed by the same rule of initiation resulting from physicochemically realistic particulars of their interaction with limiting replication firing factors.”

with:

“Our model assumes that all *p-oris* are governed by the same rule of initiation resulting from physicochemically realistic particulars of their interaction with limiting replication firing factors. […] Note however that current successful modeling of the chromosome organization of DNA replication timing relies on heterogeneities in origins’ strength and spatial distributions (Bechhoeffer and Rhind, 2012).”

3) In a related point, the authors speculate that enhanced firing rates could result from diffusion of factors released. However, there is also evidence that chromatin looping can inhibit the firing of neighboring origins. Both effects could be present, suggesting that untangling spatiotemporal correlations might be subtle.

In a recent successful modelisation of DNA replication in human, Löb et al., (2016) took into consideration both an inhibition of origin firing 55 kb around an active fork, and an enhanced firing rate further away up to a few 100 kb. However, Gindin et al., (2014) succeeded in reproducing MRT experimental profiles without introducing inhibition nor enhanced firing rate due to fork progression. These two modeling works illustrate that indeed untangling spatio-temporal correlations in replication kinetics is challenging. In that respect, 3D modeling explicitly taking into account the transport of firing factors will allow us to quantify the contribution of physicochemistry to replication spatio-temporal correlations and in turn underline the requirement for specific biological mechanisms.

In the Discussion section we have replaced:

“This opens new perspectives for understanding correlations between firing events along chromosomes that could result in part from the spatial transport of firing factors.”

with:

“Modeling of replication timing profiles in human was recently successfully achieved in a model with both inhibition of origin firing 55 kb around active forks, and an enhanced firing rate further away up to a few 100 kb (Lo¨b et al., 2016) as well as in models that do not consider any inhibition nor enhanced firing rate due to fork progression (Gindin et al., 2014; Miotto et al., 2016). These works illustrate that untangling spatio-temporal correlations in replication kinetics is challenging. 3D modeling opens new perspectives for understanding the contribution of firing factor transport to the correlations between firing events along chromosomes.”

4) When the authors modeled replication in the presence of HU, it appears that the only change made in the parameters from unperturbed replication was the speed of replication forks. Is this correct? If so, it is surprising, as activation of late-firing origins are suppressed or delayed in HU, and according to Figure 1a, one might expect less origins to be passivated with slower replication forks in HU. The authors need to comment on this.

We did not explicitly performed simulations to model replication in the presence of hydroxyurea (HU), but we simply showed that the scaling law *I_max_ ~ v ρ_0_^2^* (Eq. (5)) did apply when extracting *I_max_*and the replication fork speed *v* from data obtained in this experimental condition, keeping the same density *ρ*_0_ of *p-ori* as in normal growth condition (Confirmed, Likely and Dubious origins are taken into account).

In our model, during the second part of S-phase when most firing factors are free, the dynamics of activation of *p-oris* is controlled by the productive-interaction rate *k_on_* between a free firing factor and a *p-ori* so that a reduced replication speed will indeed result in firing of most late *p-oris* and thus a very low frequency of origin passivation. However, Alvino et al. (2007) showed that the pattern of origin firing was the same with and without HU, up to some slowing down of the progression through S-phase with HU. Up to a rescaling of time, the replication kinetics of our model is governed by the ratio between replication fork speed and *k_on_* (neglecting here the possible contribution of the activation dynamics of firing factors). Hence, our model can capture the observation of Alvino et al., (2007) considering that HU (i) induces fork slowing down and (ii) triggers a checkpoint reducing the activity of all *p-oris,* which can be modeled by a *k_on_* decrease commensurate with fork speed reduction.

In Results section we have replaced:

“Interestingly, in an experiment where T_phase_ was lengthened from 1 h to 16 h by adding hydroxyurea (HU) in yeast growth media, the pattern of activation of replication origins was shown to be conserved (Alvino et al., 2007). HU slows down the DNA synthesis to a rate of ∼ 50 bp min^−1^ corresponding to a 30 fold decrease of the fork speed (Sogo et al., 2002). In our model with a constant number of firing factors, *T_phase_ ∼ 1/vN_D_^T^*:a two fold increase of the number *N_D_^T^*of firing factors is sufficient to account for the 16 fold increase of *T_phase_*, which is thus mainly explained by the HU induced slowdown of the replication forks.”

with:

“Interestingly, in an experiment where hydroxyurea (HU) was added to the yeast growth media, the sequence of activation of replication origins was shown to be conserved even though *T_phase_*was lengthened from 1 h to 16 h (Alvino et al., 2007). HU slows down the DNA synthesis to a rate of ∼ 50 bp min^−1^ corresponding to a 30 fold decrease of the fork speed (Sogo et al., 2002). Up to a rescaling of time, the replication kinetics of our model is governed by the ratio between replication fork speed and the productive-interaction rate *k_on_*(neglecting here the possible contribution of the activation dynamics of firing factors). Hence, our model can capture the main observation of (Alvino et al., 2007) when considering a concomitant fork slowing down and *k_on_* reduction in response to HU, which is consistent with the molecular action of the replication checkpoint induced by HU (Zegerman and Diffley, 2010).”

5) Figure 2B: It was unexpected that dubious origins needed to be included for better modeling. The authors need to discuss potential reasons for this.

We agree that dubious origins are not expected to fire as frequently as Confirmed and Likely origins. A number of scenarios can contribute to the requirement for a larger number of potential origins than the number of Confirmed and Likely origins. (1) There might be some level of random initiation. (2) The maximum *I_max_*scales linearly with the fork speed. In the model we used a value of 1.5 kb/min close to the one reported in the literature (Sekedat et al., 2010; 1.6 kb/min). However, if the value of the fork speed was ∼ 2 kb/min, only Confirmed and Likely origins would be necessary to obtain the required *I_max_*value. (3) We did not consider the potential effect of the 150 *p-ori* present in the rDNA part of chromosome 12. (4).

We have replaced:

“However, in regard to the uncertainty in the value of the replication fork velocity and the possible experimental contribution of the *p-oris* in the rDNA part of chromosome 12 (not taken into account in our modeling), this conclusion needs to be confirmed in future experiments.”

with:

“This shows that in the context of our model, the number of *p-oris* required to reproduce the experimental *I(t)* curve in *S. cerevisiae* exceeds the number of Confirmed and Likely origins. Apart from the unexpected activity of Dubious origins, the requirement for a larger number of origins can be met by some level of random initiation (Czajkowsky et al., 2008) or initiation events away from mapped origins due to helicase mobility (Gros et al., 2015; Hyrien, 2016). Given the uncertainty in replication fork velocity (a higher fork speed would require only Confirmed and Likely origins) and the possible experimental contribution of the *p-oris* in the rDNA part of chromosome 12 (not taken into account in our modeling), this conclusion needs to be confirmed in future experiments.”

*6) It has been proposed that DNA replication takes place at replication foci* in vivo*, where replication factors are highly concentrated. Based on the authors' model that the localization of origins and recycling of replication factors can explain most of DNA replication kinetics, the authors need to discuss how the presence of replication foci would affect origin usage and replication kinetics.*

The modeling in this work is performed under the assumption of a well-mixed system. It thus does not address the effect of the presence of replication foci. Replication foci would locally enhance the concentration in firing factors once released from the merging of two forks. This would increase locally the kinetics in the replication foci but decrease it elsewhere. Hence, these foci are not expected to have a strong effect on the global *I(t)*. However, they are likely to induce spatial correlations in the replication program. Projection of the DNA replication program on 3D models of chromosome architecture allowed reproduction of the dynamic of replication foci in human (Löb et al., 2016). We thus expect that future work combining our model of replication kinetics with explicit modeling of firing factor 3D transport in the nucleus will allow us to address directly the nature and the consequence of replication foci on the kinetics of the replication program.

We included the following sentence at the end of the Results section:

“Inhomogeneities in origin density could create inhomogeneities in firing factor concentration that would further enhance the replication kinetics in high density regions, possibly corresponding to early replication foci.”

We also introduced the following modification to the Conclusion:

“In mammals, megabase chromosomal regions of synchronous firing were first observed a long time ago (Huberman and Riggs, 1968; Hyrien, 2016) and the projection of the replication program on 3D models of chromosome architecture was shown to reproduce the observed S-phase dynamics of replication foci (Löb et al., 2016).”

7) The paper does not cite a published model for DNA replication timing by Miotto et al., 2016 that essentially states that there are more ORC sites than are utilized during S phase and early replicating regions at the beginning of S phase is favored simply because there are far more ORC sites, whereas firing from relatively few ORC sites in late replication regions is due to increased time and the unavailability of ORC sites previously replicated. This paper should be cited and discussed to compare it to the proposed model.

We thank the reviewers for pointing this interesting article. This work reports an inhomogeneity of ORC distribution along human chromosomes, with a dense distribution of potential origins in early replicating regions (*ρ*_0*,early*_= 2.6 ORC /100 kb) and a very sparse density in late replicating regions (*ρ*_0*,late*_= 0.2 ORC /100 kb). Importantly, a model taking into account the experimental inhomogeneous distribution of ORC could account for mean replication timing profiles. The model developed in Miotto et al., (2016) does not include a limiting firing factor that controls the firing rate, instead it assumes a constant firing rate for all *p-oris*, as well as a background of random initiation. It remains unclear whether it produces a bell-shaped *I(t)* curve.

In our model, if we consider a biphasic distribution of *p-oris,* with half of the genome having a high density ρ_0,early_ and the other half a low density *ρ_0,late_*of *p-oris* with *ρ_0,earl >>_ ρ_0,late_,* most *p-oris* are located in the high density regions assuring their early replication and the origin firing kinetics (*N_fired_(t,t+dt*)) will mainly come from initiation in these regions. However, in this model, the length of unreplicated DNA also encompasses the late replicating domains resulting in a lowering of the global *I(t)* by at least a factor of 2 (Eq. (1)). Hence, in the context of our model *I_max._ 0.5vρ^2^_early_*. Interestingly, considering the experimental values for the human genome (*I_max_*= 0.3/Mb/min and v = 1.46kb/min, Table 1), it leads to ρ_0,early_ & 2.3 Ori/100 kb, in good agreement with the estimated density of 2.6 ORC/100 kb reported in this work.

We included the following at the end of Results section:

“Note that in human it was suggested that early and late replicating domains could be modeled by spatial inhomogeneity of the *p-ori* distribution along chromosomes, with a high density in early replicating domains (*ρ_0,early_*= 2.6 ORC /100 kb) and a low density in late replicating domains (*ρ_0,late_*= 0.2 ORC /100 kb) (Miotto et al., 2016). […] Hence, in the context of our model *I_max_*. 0.5vρ^2^_early_. Interestingly, considering the experimental values for the human genome (*I_max_*= 0.3/Mb/min and v = 1.46kb/min, Table 1), this leads to *ρ_0,early_*& 2.3 Ori/100 kb, in good agreement with the estimated density of 2.6 ORC/100 kb (Miotto et al., 2016).”